# LEARNING-DOMAIN DECOMPOSITION: INTERPRETING TRAINING DYNAMICS VIA LOSS VECTORS

## ABSTRACT

Deep neural networks achieve high performance, but it is still not well understood how they learn during training and when they forget what has been learned. In this study, we propose Learning-Domain Decomposition (LDD), a method that analyzes training dynamics based on per-sample loss vectors. LDD applies sparse dictionary learning to the differences of loss vectors across training steps. This enables the extraction of learning-domains, which represent common patterns learned by the model, and clarifies when they are acquired or forgotten in a bottom-up manner. We further evaluate the contribution of each domain to generalization by quantifying its effect on validation loss. Experiments on the MNIST dataset with a simple CNN show that easy samples are learned early but later degrade generalization, while ambiguous samples are repeatedly forgotten and relearned and ultimately contribute to generalization. In addition, data pruning based on the degree of contribution to multiple domains (domain multiplicity) allows training with 5% of the data while achieving performance comparable to or better than training with the full dataset. These findings demonstrate that LDD provides both an interpretable perspective on training dynamics and a practical tool for efficient data selection.

## 1 INTRODUCTION

Research on the interpretability of machine learning models has mainly focused on analyzing static, trained models (Ribeiro et al., 2016; Lundberg & Lee, 2017; Sundararajan et al., 2017; Belinkov, 2022; Huben et al., 2024). While these approaches provide valuable post-hoc explanations of model behavior, a trained model is not a static artifact but rather the result of a dynamic optimization process shaped by data. To understand how model behaviors are formed, it is essential to analyze not just the final model, but also the data and the evolution of the model's responses to that data during training. This motivates a data-centric approach to interpretability that focuses on the training dynamics. In particular, understanding which data are learned or forgotten and when this happens would also inform appropriate selection of training data (Toneva et al., 2019; Swayamdipta et al., 2020).

Such an analysis, however, presents practical challenges. Analyzing training dynamics ideally requires saving model checkpoints throughout training, but this quickly becomes prohibitive for large models. A lighter alternative is to infer dynamics from training logs such as loss and accuracy. However, training logs in machine learning typically record only per-step averages of loss and accuracy, discarding information at the level of individual samples.

We therefore analyze training dynamics via a *loss vector*, a sequence of per-sample loss calculated with fixed data samples on a specific checkpoint. The trajectory of loss vectors captures not only the evolution of overall performance but also the local behavior of the model as a function of input.

Concretely, we store the loss vector at every training step and decompose it using sparse dictionary learning (Olshausen & Field, 1996; Lee et al., 2006). We call this procedure *Learning-Domain Decomposition* (LDD). LDD identifies *learning-domains* (LDs), defined as task regions that share common patterns the model acquires. We then analyze the obtained LDs to determine when the model learns/forgets specific characteristics during training. We estimate each domain's contribution to model generalization by simulating ablations: measuring validation losses by excluding samples associated with a specific LD.

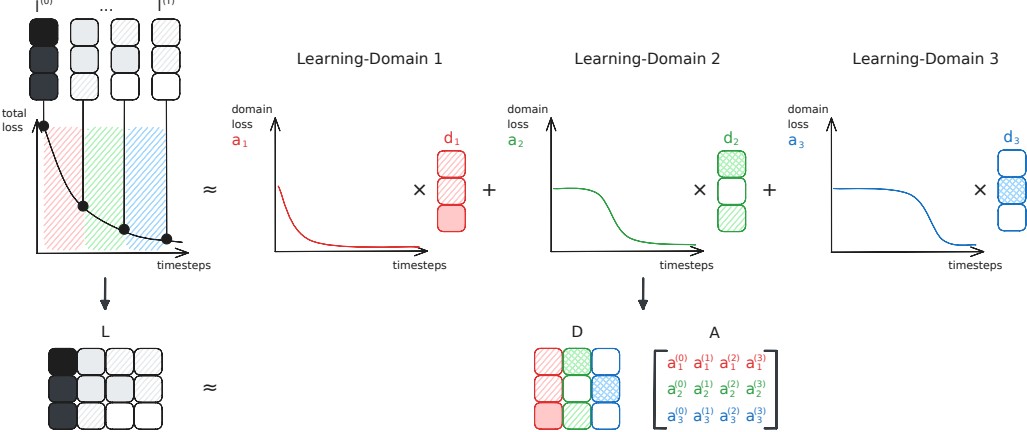

Figure 1: Conceptual overview of Learning-Domain Decomposition (LDD). We log per-sample loss vectors across training steps, form a loss matrix, and apply sparse dictionary learning to decompose it into a set of learning-domains (LDs), which are task regions that share common patterns the model acquires. Decomposed losses of LDs indicate when the model learns/forgets specific patterns.

Additionally, we investigate applicability of LDD for data selection. Based on our observation on MNIST, the model can maintain or improve its performance with fewer training examples selected to cover as many LDs with significant contribution as possible. Figure 1 shows the conceptual overview of LDD.

## 2  PRELIMINARIES

Let $\mathcal{X} := \{x_i\}_{i=1}^N$ be a pre-specified dataset (*reference set*) of sample size $N$. Denote the model parameters at optimization step $t \in \{0, 1, \ldots, T\}$ by $\theta^{(t)}$, with $\theta^{(0)}$ the initialization. The loss vector at step $t$ is defined by calculating loss values for each example in $\mathcal{X}$:

$$\boldsymbol{\ell}^{(t)} := \left[\ell(x_1; \theta^{(t)}), \ldots, \ell(x_N; \theta^{(t)})\right]^\top \in \mathbb{R}^N, \tag{1}$$

where $\ell(x; \theta)$ is the per-sample loss under parameters $\theta$. The sequence $\left(\boldsymbol{\ell}^{(0)}, \ldots, \boldsymbol{\ell}^{(T)}\right)$ evolves over training, and its trajectory reflects the model's learning dynamics observed through $\mathcal{X}$.

Collecting these column vectors yields the *loss matrix*

$$\boldsymbol{L} := [\boldsymbol{\ell}^{(0)}, \ldots, \boldsymbol{\ell}^{(T)}] \in \mathbb{R}^{N \times (T+1)}. \tag{2}$$

To emphasize changes during training, we focus on loss differences between successive steps. Define the *loss-difference vector*[1]

$$\Delta\boldsymbol{\ell}^{(t)} := \boldsymbol{\ell}^{(t-1)} - \boldsymbol{\ell}^{(t)} \in \mathbb{R}^N \quad (t = 1, \ldots, T), \tag{3}$$

and the corresponding *loss-difference matrix*

$$\boldsymbol{L}_\Delta := [\Delta\boldsymbol{\ell}^{(1)}, \ldots, \Delta\boldsymbol{\ell}^{(T)}] \in \mathbb{R}^{N \times T}. \tag{4}$$

By construction, a positive $i$-th component of $\Delta\boldsymbol{\ell}^{(t)}$ indicates that the loss on sample $x_i$ decreased at step $t$, which we interpret as learning, whereas a negative component indicates an increase of the loss, which we interpret as forgetting.

## 3  PROBLEM SETUP

Our goal is to determine (1) what the model learns/forgets and (2) when such events occurs based on a time series of loss vectors $\boldsymbol{L}$ or its difference $\boldsymbol{L}_\Delta$. We formalize the problem by separating these two questions and then unifying them.

---

[1]Note that we define $\Delta\boldsymbol{\ell}^{(t)} = \boldsymbol{\ell}^{(t-1)} - \boldsymbol{\ell}^{(t)}$, whose sign is opposite to the standard backward difference $\boldsymbol{\ell}^{(t)} - \boldsymbol{\ell}^{(t-1)}$. We adopt this convention so that positive values indicate loss decreases (learning) and negative values indicate loss increases (forgetting).

## 3.1 WHAT DOES THE MODEL LEARN?

We define a *learning-domain* (LD) as a task subregion characterized by a common pattern that the model acquires from the reference set. Samples belonging to the same domain should exhibit similar loss trajectories during training. To identify LDs, we approximate each loss vector $\ell^{(t)}$ by a linear combination of basis vectors $d_k$ ($k = 1, \ldots, K$):

$$\ell^{(t)} \approx \sum_{k=1}^{K} a_k^{(t)} d_k, \tag{5}$$

where $a_k^{(t)} \in \mathbb{R}$ are coefficients specifying the contribution of $d_k$ at the time step $t$, and $K$ is a hyperparameter determining the number of LDs. The $i$-th component of $d_k$ quantifies how strongly the sample $x_i$ contributes to the $k$-th LD. A group of high-contribution samples within an LD explains the characteristics of that LD.

Let $D := [d_1, \ldots, d_K] \in \mathbb{R}^{N \times K}$, $A := [a^{(0)}, \ldots, a^{(T)}] \in \mathbb{R}^{K \times (T+1)}$, and $a^{(t)} := [a_1^{(t)}, \ldots, a_K^{(t)}]^\top \in \mathbb{R}^K$, then we obtain the matrix notation of Equation (5):

$$\ell^{(t)} \approx D\, a^{(t)}, \qquad L \approx D\, A. \tag{6}$$

To emphasize changes, define the coefficient-difference vector $\Delta a^{(t)} := a^{(t-1)} - a^{(t)}$ for $t = 1, \ldots, T$ and $A_\Delta := [\Delta a^{(1)}, \ldots, \Delta a^{(T)}]$, then we obtain an alternative form of Equation (6):

$$\Delta\ell^{(t)} \approx D\, \Delta a^{(t)}, \qquad L_\Delta \approx D\, A_\Delta. \tag{7}$$

## 3.2 WHEN DOES THE MODEL LEARN IT?

Summing all elements of $\ell^{(t)}$ yields the total loss on the reference set at step $t$. On the other hand, each LD is expected to have its own characteristic loss curve over time, and summing these components at step $t$ should reproduce the original loss. Formally, this relationship can be represented as a constraint between $\ell^{(t)}$ and $a^{(t)}$:

$$\sum_{i=1}^{N} \ell_i^{(t)} \approx \sum_{k=1}^{K} a_k^{(t)}. \tag{8}$$

This constraint lets $a_k^{(t)}$ behave as the instantaneous loss attributable to domain $k$ at step $t$. A decrease in $a_k^{(t)}$ indicates learning of that domain, and an increase indicates forgetting.

## 3.3 JOINT FORMULATION

Given the loss-difference matrix $L_\Delta$, our problem is to recover a dictionary $D$ and a coefficient-difference matrix $A_\Delta$ such that

$$L_\Delta \approx D\, A_\Delta \quad \text{and} \quad \sum_{i=1}^{N} \ell_i^{(t)} \approx \sum_{k=1}^{K} a_k^{(t)} \text{ for all } t. \tag{9}$$

# 4 LEARNING-DOMAIN DECOMPOSITION

## 4.1 INDUCTIVE BIASES FOR INTERPRETABILITY

In addition to Equation (9), we introduce three inductive assumptions to make the decomposition identifiable and interpretable.

1. **Nonnegative contributions.** The contribution of each sample must be nonnegative: $d_{k,i} \geq 0$ for all $k, i$.

2. **Nonnegative domain losses.** The domain loss must be nonnegative: $a_k^{(t)} \geq 0$ for all $k, t$.

3. **Domain exclusivity.** When the loss associated with one domain changes substantially at a given step, other domains' losses change little. We operationalize this with sparsity in the coefficient-difference vector $\Delta a^{(t)}$.

From Assumptions 1 and 2 and the relation $\boldsymbol{\ell}^{(t)} \approx \sum_k a_k^{(t)} \boldsymbol{d}_k$ together with $\sum_i \ell_i^{(t)} \approx \sum_k a_k^{(t)}$, we obtain the (column-wise) normalization

$$\sum_{i=1}^N d_{k,i} \approx 1 \quad \text{for all } k = 1, \ldots, K. \tag{10}$$

See Appendix B for a proof.

## 4.2 Optimization

Given the loss-difference matrix $\boldsymbol{L}_\Delta$, we seek a dictionary $\boldsymbol{D} \in \mathbb{R}^{N \times K}$ and the coefficient-difference matrix $\boldsymbol{A}_\Delta \in \mathbb{R}^{K \times T}$ that satisfy the factorization $\boldsymbol{L}_\Delta \approx \boldsymbol{D} \boldsymbol{A}_\Delta$ while enforcing non-negativity on $\boldsymbol{D}$, simplex normalization of its columns, and sparsity on $\boldsymbol{A}_\Delta$:

$$
\begin{aligned}
\min_{\boldsymbol{D}, \boldsymbol{A}_\Delta} \quad & \|\boldsymbol{L}_\Delta - \boldsymbol{D} \boldsymbol{A}_\Delta\|_F^2 + \lambda \|\boldsymbol{A}_\Delta\|_{1,1} \\
\text{s.t.} \quad & \boldsymbol{D} \geq 0, \qquad \mathbf{1}^\top \boldsymbol{D} = \mathbf{1}^\top,
\end{aligned}
\tag{11}
$$

where $\|\cdot\|_F$ is the Frobenius norm, $\|\cdot\|_{1,1}$ is the sum of absolute values, $\lambda > 0$ is a regularization parameter, and $\mathbf{1}$ is the all-ones vector (so each column of $\boldsymbol{D}$ sums to 1). The $\ell_1$ penalty encourages temporal sparsity in $\boldsymbol{A}_\Delta$, aligning with domain exclusivity in Assumption 3. We estimate $(\boldsymbol{D}, \boldsymbol{A}_\Delta)$ via alternating minimization: (i) solve a sparse coding subproblem for $\boldsymbol{A}_\Delta$ with $\boldsymbol{D}$ fixed; (ii) update $\boldsymbol{D}$ with nonnegativity and simplex projections; iterate to convergence. In practice, we implement this with scikit-learn's `DictionaryLearning`.[2]

This yields a set of basis vectors $\{\boldsymbol{d}_k\}$ (nonnegative, sample-weighted patterns) and a coefficient-difference time series $\{\Delta \boldsymbol{a}^{(t)}\}$. Integrating the differences recovers $\boldsymbol{a}^{(t)}$ up to a constant, which we fix to satisfy $\sum_i \ell_i^{(t)} \approx \sum_k a_k^{(t)}$.

## 5 Experiments

### 5.1 Experimental setup

**Dataset.** We evaluate on MNIST (LeCun et al., 1998) (handwritten digit recognition). The training set contains 30,000 examples and the validation set 3,000 examples. We use the training dataset as the reference set for computing loss vectors.

**Model.** We use a simple convolutional neural network (CNN) with two convolutional layers followed by a fully connected layer. The first convolutional layer has 32 output channels, the second has 64 channels, and the fully connected layer has a hidden size of 128.

**Training.** We minimize cross-entropy loss using Adam (Kingma & Ba, 2015) with learning rate $1 \times 10^{-4}$. Training is run for a single epoch, corresponding to 100 optimization steps (30,000 samples / batch size 300). We record the full per-sample loss vector at every step.

**Learning-Domain Decomposition (LDD).** For the dictionary factorization of loss differences, we set the number of domains $K = 10$ and regularization $\lambda = 0.01$.

**Data pruning.** We retrain models on pruned subsets using the same optimization hyperparameters as above. To ensure comparability, the number of optimization steps is kept the same after pruning by adjusting the number of epochs.

### 5.2 Visualizing learning-domains

The learned ten basis vectors $\boldsymbol{d}_1, \ldots, \boldsymbol{d}_{10}$ each contain 30,000 nonnegative entries. By examining which training samples receive large values within each basis vector, we can identify the data samples associated with each learning-domain.

---

[2]https://scikit-learn.org/stable/modules/generated/sklearn.decomposition.DictionaryLearning.html

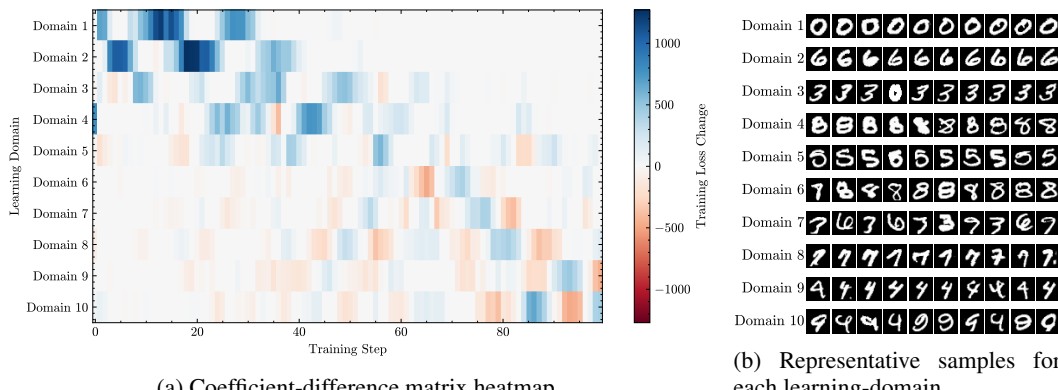

(a) Coefficient-difference matrix heatmap.

(b) Representative samples for each learning-domain.

Figure 2: Learning-domains discovered by LDD. (a) Coefficient-difference matrix heatmap ($A_\triangle$). The color of each cell encodes the decrease in domain loss at that step, $\Delta a_k^{(t)}$. Blue cells indicate that the loss for that domain decreased ($\Delta a_k^{(t)} > 0$), whereas red cells indicate that it increased ($\Delta a_k^{(t)} < 0$). (b) Representative samples for each learning-domain. We display the 10 training images with the highest basis coefficients $d_{k,i}$ for each domain $k$. These samples show the characteristic patterns that define each domain.

Figure 2a shows a heatmap of the coefficient-difference matrix $A_\triangle$. This heatmap visualizes domain-specific patterns of loss change. For example, domains 1 and 2 show concentrated blue early in training, indicating that the model substantially improves those domains at the beginning. In contrast, domains 9 and 10 alternate between blue and red, indicating repeated cycles of loss decrease (learning) and increase (forgetting).

Figure 2b presents examples of the data samples that contribute most to each domain. Specifically, for each basis vector $d_k$, we display the images corresponding to the top 10 entries with the largest values. These can be interpreted as representative examples of the data primarily covered by that learning-domain. The representative images exhibit characteristic structure across domains. For instance, domain 1 contains many images of the digit "0", whereas domain 10 mixes visually confusable digits such as "4" and "9". This indicates that the learned basis vectors detect semantically coherent clusters in the training data.

Comparing the heatmap (Figure 2a) with the representative images (Figure 2b) allows us to interpret when each learning-domain is acquired. Domains with large early loss drops (e.g., domains 1 and 2) tend to contain relatively simple, easily identifiable digits, aligning with the intuition that the model learns easy patterns first. In contrast, domains whose losses decrease only later and fluctuate with repeated increases and decreases (e.g., domains 5 and 10) include many confusable pairs such as "5" vs. "8" and "4" vs. "9". The model may learn them temporarily during training, but when learning other patterns and adjusting the decision boundaries, it can misclassify them again.

To assess the discovered learning-domains globally, we visualize the per-sample loss-change trajectories with t-SNE. Coloring points by ground-truth digit labels yields well-separated clusters (Figure 3). When we instead visualize per-domain contributions in the same embedding, each domain concentrates on a largely non-overlapping region, exhibiting mutually exclusive distributions (Figure 4).

### 5.3 DOMAIN-WISE CONTRIBUTIONS TO GENERALIZATION

To understand how each discovered learning-domain contributes to the model's generalization ability, we perform a counterfactual analysis. For each training step $t$, we first compute the validation loss using the current model $\theta^{(t)}$. We then simulate what would happen if we excluded samples from a specific domain $k$ during the update from step $t$ to $t+1$. Specifically, we remove training examples that have large coefficients in the corresponding basis vector $d_k$ from the mini-batch. By comparing the resulting validation loss change to the original update, we can estimate how much domain $k$ contributes to generalization. When excluding domain $k$ leads to worse validation performance,

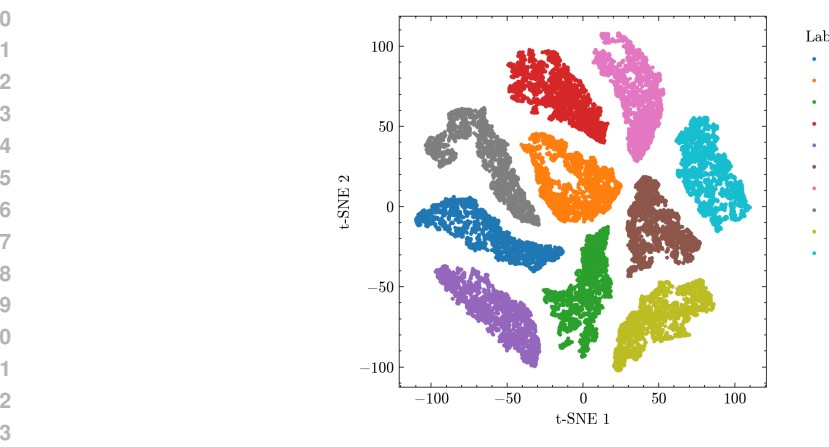

Figure 3: t-SNE of per-sample loss-change trajectories. Each point corresponds to a training example embedded from its sequence of $\Delta\ell$ values. Color denotes the true digit label.

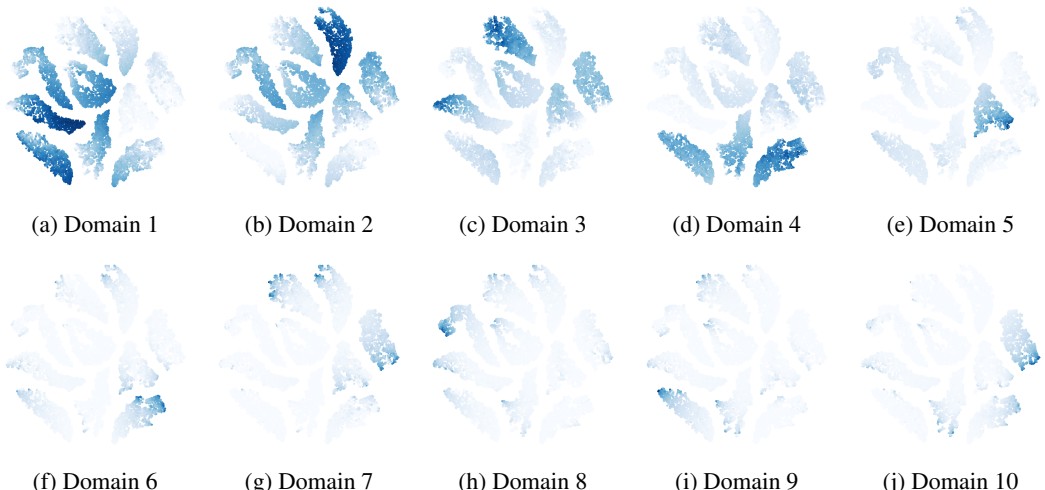

| (a) Domain 1 | (b) Domain 2 | (c) Domain 3 | (d) Domain 4 | (e) Domain 5 |

| (f) Domain 6 | (g) Domain 7 | (h) Domain 8 | (i) Domain 9 | (j) Domain 10 |

Figure 4: Distribution of domain contributions in t-SNE space. For each domain $k$, color intensity indicates the domain contribution $d_{k,i}$ for sample $x_i$ normalized by the maximum and minimum values within each domain. Brighter means larger contribution.

we interpret this as evidence that domain $k$ positively contributes to generalization. Conversely, if removing domain $k$ actually improves validation performance, we conclude that this domain may be harmful to generalization.

Figure 5 shows how these domain contributions evolve throughout training. For each step, the vertical axis represents the difference between validation loss after updating without domain $k$ versus the original validation loss. Red regions indicate that excluding the domain would degrade performance (positive contribution), while blue regions suggest excluding the domain would improve performance (negative contribution). Interestingly, we observe that domains 5 and 10 consistently provide substantial positive contributions across training, suggesting these more challenging domains play a crucial role in improving generalization. Meanwhile, the simpler domains that are learned early (such as domains 1 and 2) initially contribute positively but later become detrimental, implying that overemphasizing these easy patterns may eventually interfere with generalization.

## 5.4 APPLICATION TO DATA PRUNING

Assumption 3 implies that samples strongly associated with the same domain tend to improve together. Learning one often reduces the loss of many others in that domain. We quantify each

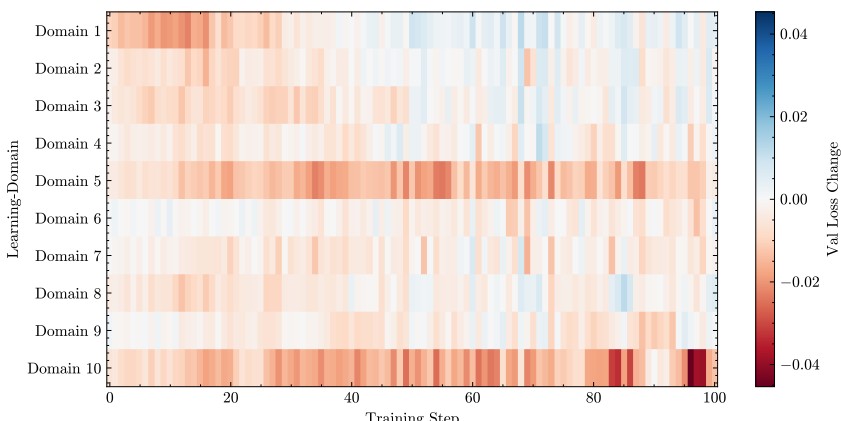

Figure 5: Domain-wise contributions to validation loss over training steps. Red indicates a positive contribution, while blue indicates a negative contribution.

sample's domain multiplicity

$$m_i \;:=\; \sum_{k=1}^{K} d_{k,i}. \tag{12}$$

Intuitively, $m_i$ measures how broadly sample $x_i$ participates across domains. We normalize $\{m_i\}$ to a probability distribution and prioritize samples with larger $m_i$ when constructing a reduced training set. Concretely, to prune from size $N$ to $n < N$, we sample without replacement $n$ examples according to probabilities proportional to $m_i$, and retrain under the same protocol. This coverage-based criterion favors examples that simultaneously advance multiple learning-domains. We compare against random pruning.

**Varying retained data size (fixed $K = 10$).** Figure 6a shows test accuracy as a function of the retained fraction. Retaining as little as 2% of the data outperforms random pruning at the same budget, and with $\geq 5\%$ retained, the pruned model matches or exceeds the accuracy of training on the full dataset.

**Varying the number of domains (fixed $n = N/10$).** Figure 6b fixes the retained fraction at 10% and varies $K$. When $K \leq 2$, accuracy falls below random pruning, suggesting under-decomposition. For $K \geq 10$, LDD-based pruning consistently outperforms random pruning. This indicates that, for pruning, the dictionary should be sufficiently expressive to capture multiple, distinct learning-domains.

## 6 DISCUSSION

### 6.1 DISTRIBUTION OF SAMPLE IMPORTANCE AS A FUNCTION OF THE NUMBER OF DOMAINS

Figure 7 visualizes a two-dimensional t-SNE embedding of the per-step loss-change vectors, where points correspond to training samples and color denotes their importance under Learning-Domain Decomposition (LDD). Importance here refers to the domain multiplicity $m_i = \sum_{k=1}^{K} d_{k,i}$, which determines a sample's retention probability in our pruning scheme.

Across settings $K \in \{1, 2, 5, 10, 20, 50, 100\}$, we observe that once the dictionary has sufficient capacity ($K \geq 10$), the spatial pattern of importance stabilizes. The high-importance regions and their relative extents change little as $K$ increases further. This agrees with the pruning results (Figure 6b), where $K \geq 10$ consistently outperforms random pruning, while very small $K$ (e.g., $K \leq 2$) under-decomposes the dynamics and degrades accuracy.

A common geometric trend is that, near convergence, importance concentrates along cluster perimeters in the t-SNE map, that is, near putative decision boundaries. Samples on these margins appear

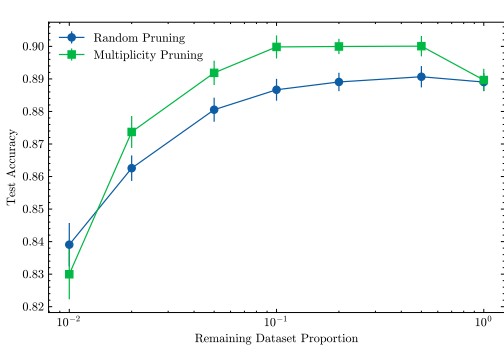

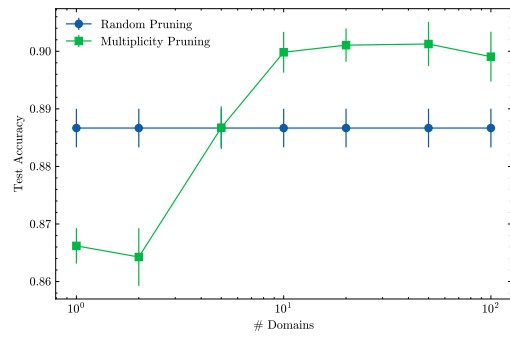

(a) Test accuracy vs. retained data fraction ($K = 10$).

(b) Test accuracy vs. number of domains (10% retained).

Figure 6: Test accuracy after LDD-based data pruning. (a) Test accuracy as a function of retained data fraction with $K = 10$ domains. (b) Test accuracy as a function of the number of domains with 10% of data retained. Error bars denote 95% confidence intervals across 10 runs.

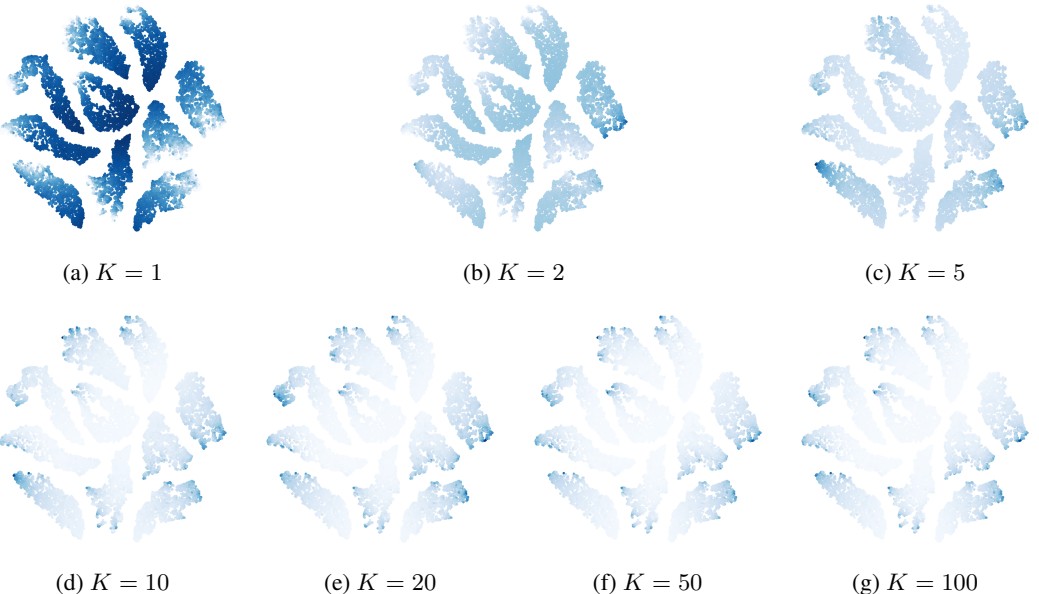

(a) $K = 1$     (b) $K = 2$     (c) $K = 5$

(d) $K = 10$    (e) $K = 20$    (f) $K = 50$    (g) $K = 100$

Figure 7: Sample importance $m_i$ in t-SNE space for varying $K$.

to participate in multiple learning-domains (higher $m_i$), so training on them tends to advance several domains simultaneously. Consequently, although our criterion is derived from coverage in domain space (not from predictive uncertainty per se), the selected subsets qualitatively resemble those produced by uncertainty sampling, providing an intuitive link between domain coverage and decision-boundary refinement.

## 7 RELATED WORK

**Loss vectors.** Per-sample loss (or log-likelihood) vectors have been used as features of model behavior. For language models, prior work observes that KL divergence between models can be approximated from negative loss (log-likelihood) vectors, and that such vectors serve as useful features for downstream analysis. In particular, log-likelihood vectors have recently been shown to be informative representations of language models themselves (Oyama et al., 2025b;a). Despite this promise, the use of per-sample loss vectors to analyze training dynamics, rather than to compare fixed models, remains underexplored.

**Visualizing training dynamics.** Several studies, akin to our perspective, visualize trajectories of loss vectors (or related statistics) via dimensionality reduction, e.g., ISOMAP (Erhan et al., 2010), PCA (Olsson et al., 2022), and t-SNE (Kishino et al., 2025). These works explain global properties, such as the stability of pretraining or phase transitions during learning, by interpreting the shape of the trajectory. However, the resulting coordinates are not inherently interpretable, so such visualizations typically do not reveal which data are being learned or when specific patterns are acquired or forgotten.

**Top-down identification of acquired knowledge.** A complementary, top-down line of work investigates concrete skills that emerge during training. For example, Chen et al. report that syntactic attention structures in masked language models appear abruptly at particular training stages, aligning with sharp drops in loss (i.e., phase transitions) (Chen et al., 2024). These approaches project learning dynamics onto predefined, semantically labeled features. This elucidates the emergence of targeted capabilities. In contrast, our method is bottom-up. From only the time series of loss vectors, we automatically extract nonnegative, sparse bases (learning-domains). We jointly identify when they are learned and which samples define them, without requiring prior semantic annotations. In this way, our approach complements top-down analyses.

**Data pruning.** Data pruning aims to curb training cost while preserving accuracy. One influential family estimates example importance from early-training behavior, including heuristics such as GraNd (Paul et al., 2021) and EL2N (Paul et al., 2021), which provide quick proxies for data utility. Sorscher et al. (Sorscher et al., 2022) further argue that such pruning signals can improve the scaling of accuracy with dataset size. A second family frames pruning as subset selection via gradient matching or bilevel optimization, as in CRAIG (Mirzasoleiman et al., 2019), GRAD-MATCH (Killamsetty et al., 2021), and GLISTER (Killamsetty et al., 2020), which offer principled objectives but can be computationally demanding and may depend on a validation set. Other work estimates per-example value through influence functions (Koh & Liang, 2017) or Shapley values (Ghorbani & Zou, 2019), which are theoretically grounded yet often expensive to compute at scale. Orthogonally, studies of forgetting events (Toneva et al., 2019) and confidence trajectories (Swayamdipta et al., 2020) show that ambiguous samples can aid generalization, inspiring diagnostic tools such as Data Maps (Swayamdipta et al., 2020). Our approach differs by decomposing the time series of loss vectors into learning-domains and measuring a sample's value by its coverage, defined as the number of domains it advances (domain multiplicity), rather than by difficulty alone.

## 8 CONCLUSION

We introduced a data-driven, bottom-up method for analyzing the training dynamics of deep neural networks. By decomposing per-step loss vectors into nonnegative, sparse basis vectors (Learning-Domain Decomposition, LDD), our framework jointly identifies what is learned (learning-domains) and when it is learned or forgotten.

On MNIST with a simple CNN, LDD reveals: (i) easy patterns are acquired early, while ambiguous patterns undergo cycles of forgetting and relearning; (ii) these ambiguous domains make sustained, positive contributions to generalization, whereas easy domains can become detrimental later in training. Leveraging these insights, we formulate a coverage-based pruning strategy using domain multiplicity, which maintains accuracy and, in some settings, improves it, while substantially reducing the training set.

Future work includes scaling LDD to larger architectures and datasets, extending it to language models, and exploring applications to post-training regimes (e.g., instruction tuning, reinforcement learning).

ETHICS STATEMENT

Our study analyzes training dynamics on the publicly available MNIST dataset and does not involve human subjects, personally identifiable information, or sensitive attributes.

REPRODUCIBILITY STATEMENT

Experimental settings, including dataset, network architectures, optimization parameters, and LDD hyperparameters (e.g., number of domains $K$, sparsity constraints), are described in Section 5.1.

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

# A  USE OF LARGE LANGUAGE MODELS

The authors used large language models (ChatGPT[3] and Gemini[4]) for translation, grammar correction, help finding related work, and writing a draft of the abstract. Outputs have been reviewed and edited by the authors.

# B  PROOF OF COLUMN-WISE NORMALIZATION OF $D$

We justify the relation

$$\sum_{i=1}^{N} d_{k,i} \approx 1 \quad \text{for all } k = 1, \ldots, K. \tag{13}$$

Recall from the main text that for each optimization step $t$ we have the (approximate) nonnegative decomposition

$$\boldsymbol{\ell}^{(t)} \approx \sum_{k=1}^{K} a_k^{(t)} \, \boldsymbol{d}_k \;=\; \boldsymbol{D} \, \boldsymbol{a}^{(t)} \tag{14}$$

with $\boldsymbol{D} \geq 0$ (Assumption 1) and $\boldsymbol{a}^{(t)} \geq 0$ (Assumption 2). We also impose the identifiability convention that the total loss is approximated by the sum of the domain coefficients,

$$\sum_{i=1}^{N} \ell_i^{(t)} \;\approx\; \sum_{k=1}^{K} a_k^{(t)} \;=\; \mathbf{1}^{\top} \boldsymbol{a}^{(t)} \quad \text{for all } t. \tag{15}$$

Summing the decomposition over samples and using linearity gives

$$\sum_{i=1}^{N} \ell_i^{(t)} \;\approx\; \sum_{k=1}^{K} a_k^{(t)} \left( \sum_{i=1}^{N} d_{k,i} \right) \;=\; \mathbf{1}^{\top} \boldsymbol{D} \, \boldsymbol{a}^{(t)}. \tag{16}$$

Comparing the two expressions for $\sum_i \ell_i^{(t)}$ yields

$$\mathbf{1}^{\top} \boldsymbol{D} \, \boldsymbol{a}^{(t)} \;\approx\; \mathbf{1}^{\top} \boldsymbol{a}^{(t)} \quad \text{for all } t. \tag{17}$$

Because both sides are linear in $\boldsymbol{a}^{(t)}$ and the coefficients $\boldsymbol{a}^{(t)}$ vary over training (and are nonnegative but not identically zero), the only way for this to hold for all $t$ is

$$\mathbf{1}^{\top} \boldsymbol{D} \;\approx\; \mathbf{1}^{\top}, \tag{18}$$

that is, each column of $\boldsymbol{D}$ approximately sums to 1:

$$\sum_{i=1}^{N} d_{k,i} \;\approx\; 1 \quad (k = 1, \ldots, K). \tag{19}$$

In the idealized, noise-free case where the equalities above hold exactly and the set $\{\boldsymbol{a}^{(t)}\}_t$ spans $\mathbb{R}^K$, the conclusion strengthens to $\mathbf{1}^{\top} \boldsymbol{D} = \mathbf{1}^{\top}$. In practice we enforce this column-simplex constraint during optimization (via projection), making the normalization exact by design.

---

[3]https://chatgpt.com/
[4]https://gemini.google.com/app

