# OpenReview forum: "Learning-Domain Decomposition: Interpreting Training Dynamics via Loss Vectors"
_ICLR.cc/2026/Conference — ICLR 2026 Conference Withdrawn Submission_

### Official Review · Reviewer_dkDv · 2025-10-30

**Soundness:** 1
**Presentation:** 3
**Contribution:** 2
**Rating:** 2
**Confidence:** 4

**Summary:**

The paper introduces Learning-Domain Decomposition (LDD), a framework to interpret the training dynamics of neural networks through per-sample loss vectors. LDD tracks how losses for each sample evolve over time by forming loss-difference matrix decomposed via sparse dictionary learning. Each basis vector corresponds to a learning-domain (LD)—a coherent subset of data exhibiting similar learning·forgetting patterns. Experiment on MNIST reveals interpretable trends that easy samples are learned early but later hurt generalization, and ambiguous samples are repeatedly forgotten and relearned. LDD also yields a data pruning heuristic based on domain multiplicity, reducing the dataset to 5% without severe performance degradation.

**Strengths:**

- The paper presents an novel and interesting framework that interprets neural network training dynamics by decomposing per-sample loss trajectories through sparse dictionary learning.
- The mathematical formulation is clear, and its formulation of the imposed nonnegativity and sparsity constraints lead to interpretable learning domains.

**Weaknesses:**

- The experimental validation is limited to MNIST and a small CNN, leaving scalability to larger datasets untested.
- The paper does not compare LDD with existing baselines, making its relative contribution unclear.
- Ablations on key hyperparameters such as K and λ are minimal.

**Questions:**

- How stable are the domains under different seeds or batch orders?

---

### Official Review · Reviewer_R5CJ · 2025-10-31

**Soundness:** 2
**Presentation:** 2
**Contribution:** 2
**Rating:** 2
**Confidence:** 5

**Summary:**

This paper introduces Learning-Domain Decomposition (LDD), a method for interpreting the training dynamics of deep neural networks by analyzing the evolution of per-sample loss vectors. The technique applies sparse dictionary learning to the sequence of loss differences across training steps, extracting "learning-domains" (LDs) that purportedly capture shared patterns learned or forgotten during training. The method is evaluated on MNIST with a simple CNN, showcasing how LDs may inform generalization, guide pruning, and yield insight into the order and stability of learned representations.

**Strengths:**

- The paper addresses a timely and underexplored problem: interpretable analysis of neural network training dynamics from a data-centric perspective, focusing on per-sample loss vectors.
- The proposed LDD framework is conceptually clear and grounded in established sparse coding and matrix factorization methodologies. The assumptions and constraints (nonnegativity, sparsity, column normalization) are made explicit, and their interpretability motivation is cogently discussed.
- The presentation of the method's pipeline and concepts is visually clear—especially in Figures 1 and 2—which help in understanding how LDs are derived from loss dynamics and associated with exemplar samples.

**Weaknesses:**

1. The entirety of the experimentation is based on MNIST with a basic CNN. This is a very limited test and bed for a method that purports to reveal general insights about neural network learning dynamics. Extending experiments to more challenging benchmarks such as CIFAR-100 or ImageNet would substantially strengthen the paper’s empirical foundation.
2. The need to store per-step, per-sample loss vectors for thousands of examples across all steps (e.g., entire MNIST) could quickly become infeasible for larger datasets or models.
3. LDD relies on sparse dictionary learning to factorize the loss-difference matrix L_\Delta=DA_\Delta. However, such factorization is generally non-unique and sensitive to random initialization, regularization strength (\lambda), and the number of domains K. The paper lacks a systematic analysis of how these hyperparameters or random seeds affect the resulting learning-domains and their interpretability.
4. The data pruning results are only compared against random selection, without benchmarking against other existing pruning or data selection methods (e.g., GraNd, EL2N, CRAIG, or GradMatch).

**Questions:**

The same as weaknesses.

**Details Of Ethics Concerns:**

N/A.

---

### Official Review · Reviewer_vz4f · 2025-11-05

**Soundness:** 2
**Presentation:** 3
**Contribution:** 2
**Rating:** 2
**Confidence:** 4

**Summary:**

The paper proposes a method for analysing training dynamics. Their approach clusters examples based on their training dynamics into learning domains, viz. pace of learning, learning unlearning, etc. Learning domains reveal what and when is learned during training, which may improve the understanding of a developer.

**Strengths:**

Making sense of training dynamics to interpret problematic or impactful data subsets is an important and topical problem.

**Weaknesses:**

The contributions are not crisp. As I see, the paper makes the following contributions: (a) an analysis tool that brings out the various domains based on their learning dynamics, (b) data selector which matches full data accuracy with only 5% data, (c) an insight based on MNIST experiment that ambiguous inputs such as 4, 9 are consistently important for performance.

All the contributions (a, b, c) are not convincingly supported by the evidence and experiments presented in the paper.
For (a), further evidence is required to show that automatically discovered domains are interpretable to the user when presented with the top-k elements affiliated with that domain.
 For (b), the motivation for data selection is unclear but even if their approach has merits in non-trivial data selection, we need to see validation beyond MNIST.
For (c), the discovered domains in fig-2-b are aligned with different classes. Therefore, dropping domain 10 in the figure could be dropping majority of label '9' examples, which could lead to worser validation accuracy due to the induced label bias. Therefore, the insight that ambiguous examples are important for learning is not the right takeaway. In any case, if (c) is a primary contribution, we need better validation.

I am not fully aware of the literature, but some very relevant references are missing such as [1]

References.
[1] https://arxiv.org/pdf/2209.10015

**Questions:**

Please address the concerns raised in the "Weaknesses" section.

---

### Note · Authors · 2025-11-16

**Comment:**

We would like to thank all reviewers for the time and effort you invested in evaluating our work. After considering your feedback, we have decided to withdraw the paper so that we can further develop this research. We appreciate your comments, which will be helpful as we improve our study.

**Withdrawal Confirmation:**

I have read and agree with the venue's withdrawal policy on behalf of myself and my co-authors.